# Gain-Framed Health Messaging, Medical Trust, and Pre-Exposure Prophylaxis (PrEP) Self-Efficacy: An Experimental Study

**DOI:** 10.3390/healthcare13161981

**Published:** 2025-08-12

**Authors:** Anthony J. Gifford, Rusi Jaspal, Bethany A. Jones, Daragh T. McDermott

**Affiliations:** 1NTU Psychology, School of Social Sciences, Nottingham Trent University, Nottingham NG1 4FQ, UK; beth.jones@ntu.ac.uk (B.A.J.); daragh.mcdermott@ntu.ac.uk (D.T.M.); 2Vice Chancellor’s Office, University of Brighton, Brighton BN2 4G, UK; r.jaspal@brighton.ac.uk

**Keywords:** pre-exposure prophylaxis (PrEP), self-efficacy, medical mistrust, identity resilience, gain-framed messaging

## Abstract

**Background:** Despite the clinical efficacy of pre-exposure prophylaxis (PrEP) in preventing HIV, uptake remains suboptimal among men who have sex with men (MSM) in the United Kingdom (UK). Sustaining progress in the PrEP cascade requires more than biomedical availability; it demands effective, psychologically informed interventions that address persistent barriers. Psychological factors, such as medical mistrust, low PrEP self-efficacy, and identity-related processes, continue to undermine engagement. This study tested whether narrative persuasion and message framing could influence these barriers. **Method:** A sample of 253 MSM was recruited to participate in an online experiment and completed baseline measures of identity resilience before being randomly allocated to either the gain-framed (*N* = 122) or loss-framed (*N* = 124) narrative condition and then completing post-manipulation measures of medical mistrust and PrEP self-efficacy. After excluding 7 cases due to ineligibility, data from 246 participants were analysed using mediation analysis. **Results:** Participants in the gain-framed condition reported lower medical mistrust than those in the loss-framed condition. Medical mistrust was, in turn, associated with lower PrEP self-efficacy. Identity resilience was associated with lower medical mistrust and higher PrEP self-efficacy. **Discussion:** These findings provide novel causal evidence that gain-framed health narratives can reduce mistrust and indirectly enhance PrEP self-efficacy. Identity resilience also emerges as a key psychological factor influencing trust and behavioural confidence. **Conclusions:** Interventions to improve and sustain PrEP uptake should combine gain-framed, narrative-based messaging with strategies to bolster identity resilience. Such approaches may address psychosocial barriers more effectively and promote equitable PrEP uptake among MSM.

## 1. Introduction

Pre-exposure prophylaxis (PrEP) is a highly effective biomedical intervention for the prevention of human immunodeficiency virus (HIV) acquisition [1]. Despite excellent clinical efficacy, PrEP uptake is stagnating among key populations at elevated risk of HIV acquisition, such as men who have sex with men (MSM) [2]. Furthermore, while rates of new HIV infections in MSM are decreasing in the United Kingdom (UK), this decline is starting to plateau [3]. Increasing PrEP uptake is crucial for achieving the UNAIDS [4] target of no new HIV infections by 2030. Thus, there is a pressing need to identify and test innovative approaches to increase PrEP uptake that address both psychological and structural barriers [5,6].

Golub and colleagues [7] state that psychological variables, such as PrEP self-efficacy (i.e., confidence in one’s ability to access and consistently take PrEP as prescribed), influence decision-making in relation to PrEP uptake. Low PrEP self-efficacy, shaped by barriers such as financial cost and time commitment, can undermine people’s ability to initiate PrEP, highlighting the need for interventions that enhance self-efficacy [8]. Furthermore, Jaspal [9] argues that fostering trust in medicine and science and reducing stigma are essential for promoting willingness to take PrEP. Indeed, medical mistrust, defined as suspicion of or lack of confidence in healthcare systems, providers, or medical information, has been linked to lower perceived PrEP self-efficacy [10,11]. Accordingly, to increase PrEP uptake, it is crucial to develop communication strategies and educational interventions that not only provide accurate information about PrEP but also build users’ trust in healthcare systems, address stigma, and improve PrEP self-efficacy.

One promising strategy for achieving these goals is effective health communication. Narrative persuasion, in particular, is regarded as a powerful strategy for engaging marginalized groups with health messages [12,13,14]. Unlike traditional informational messages or fear appeals, narrative persuasion communicates health content through storytelling-using characters, plots, and scenarios that resonate with the target audience. For example, Ma and colleagues [15] used the narrative of *Jennifer* to communicate risks of alcohol consumption and breast cancer, and Kim and colleagues [16] used a first-person narrative about receiving results from a human papilloma virus (HPV) test. This study sought to develop a narrative about PrEP uptake that was relatable to the target population (i.e., MSM) by presenting a character that navigates PrEP decisions, dramatized through lived experiences that communicate risks and benefits in a more engaging and accessible way than didactic instruction. This approach has a theoretical advantage as analogous research shows that well-crafted narratives can transport viewers into the story and foster identification with characters, which in turn reduces counter-arguing and psychological resistance to the health promotion message [17].

Within public health, narrative persuasion has demonstrated unique efficacy in reducing resistance to counter-attitudinal messages, such as vaccine hesitancy. For example, Nan and colleagues [18] found that participants exposed to a narrative vignette reported higher perceived risk of HPV and greater intentions to receive the vaccine compared to those who received non-narrative messages. Furthermore, Murphy and colleagues [19] found that a fictional narrative film about cervical cancer led to greater screening uptake than a non-narrative video, with reportedly greater knowledge and more positive attitudes towards a “Pap test”. Finally, in their anti-smoking study, Igartua and Rodríguez-Contreras [20] found that first-person narrative testimonials led to greater identification with the protagonist and that this emotional connection strengthened the persuasive impact of the message, increasing intentions to quit smoking. These findings position narrative messaging as a widely used and highly impactful strategy across public health interventions, particularly in engaging resistant audiences and promoting behaviour change.

Another important consideration in health narratives is the way in which the risks and benefits of health behaviours are “framed” in subsequent communications [21]. While a narrative is important, key messages in terms of gains or losses related to potential outcomes can also significantly influence behaviour. In health communication, a gain-framed message emphasises the benefits of taking action, whereas a loss-framed message highlights the costs of not doing so [22]. For example, a gain-framed PrEP message might stress that “by taking PrEP, you stay HIV-negative and protect your health” while a loss-framed equivalent might warn that “if you don’t take PrEP, you risk contracting HIV and harming your health.” Typically, loss-framed messaging has been found to be more effective for promoting reactive health behaviours, as it heightens perceived vulnerability and motivates action [23]. However, this tends to be less effective for preventive behaviours and, in the context of PrEP specifically, the relationship is complex. The paradox in relation to PrEP is that messaging that focuses on risk or disease can inadvertently reinforce stigma, evoke defensiveness, or conflict with identities [24,25].

Recent studies indicate, instead, that gain-framed messaging, focused on empowerment, control, and sexual wellbeing, is more effective in promoting PrEP acceptability among MSM [26]. After all, such framing aligns with identity-affirming narratives and reduces the perception of PrEP as a marker of elevated risk or moral deviance. However, this does not preclude the use of loss-framed messages altogether. Rather, it suggests that loss-framed messages with a positive valence may offer a more effective alternative. For example, framing non-use of PrEP as a missed opportunity to take control of one’s health, rather than as a moral failure or danger. In this way, loss can be framed in terms of lost empowerment or peace of mind, rather than fear-based consequences. Such an approach may strike a balance between urgency and affirmation, motivating behaviour change without exacerbating stigma or identity threat.

Furthermore, recent research by Gifford and colleagues [27] extends this perspective by demonstrating that identity resilience (i.e., a construct comprising self-esteem, self-efficacy, continuity, and positive distinctiveness) predicts domain-specific outcomes, such as PrEP self-efficacy. It reflects the capacity to maintain a stable, valued sense of self when confronted with threats, such as PrEP stigma [28], and has demonstrated clear utility in health behaviour contexts. For example, Breakwell [29] found that when people were primed to think about the COVID-19 pandemic, those with higher identity resilience experienced significantly less fear arousal. Likewise, identity resilience can shape vaccine uptake attitudes, given that components of identity resilience have been found to be associated with more positive views of COVID-19 vaccines and a greater likelihood of vaccination [30]. Building on these insights, Gifford and colleagues [27] used the concept in the context of HIV prevention and showed that identity resilience plays a key role in determining PrEP use. In their study of MSM in the UK, higher identity resilience significantly predicted greater PrEP self-efficacy. Furthermore, identity resilience was linked to greater trust in science and lower PrEP stigma. Thus, independently of traditional determinants of PrEP use, such as HIV risk perception [31], psychological constructs relating to identity appear to drive PrEP uptake. Therefore, interventions for improving PrEP uptake should not only target PrEP self-efficacy directly but also incorporate identity resilience.

Despite extensive research identifying barriers to PrEP uptake and adherence (e.g., [5,28,29]), there remains a paucity of research into effective strategies to improve PrEP uptake. This study addresses this lacuna. It has two main aims: first, to examine the utility of narrative persuasion and loss/gain-framed messaging in reducing medical mistrust and enhancing PrEP self-efficacy; and, second, to examine the role of identity resilience in relation to both trust in medicine and PrEP self-efficacy. The results are of particular importance for MSM who are not currently using PrEP but who may benefit from it due to their HIV risk profile [32,33]. They should be used to develop future health interventions for increasing PrEP uptake.

To assess the potential of narrative persuasion, culturally relevant health narratives featuring MSM protagonists were developed to reduce feelings of alienation or mistrust that generic campaigns may provoke [24,34,35]. Loss/gain-framed messaging was experimentally manipulated to examine its causal impact on PrEP self-efficacy. Finally, identity resilience was included as a covariate, given its theoretical relevance to both medical mistrust and PrEP self-efficacy. This mediation model proposes that message framing influences PrEP self-efficacy via medical mistrust, as loss-framed messages may heighten medical mistrust among MSM, reducing confidence in PrEP use [10,36,37]. Medical mistrust is a known barrier to PrEP engagement, and the concept of identity resilience was included in the model given its demonstrated role in buffering the effects of mistrust and reinforcing self-efficacy, particularly in the face of identity-relevant threats, such as stigma or mistrust in healthcare [28,38].

The following specific hypotheses are tested:
**H1.** *There will be a statistically significant difference in levels of medical mistrust between participants exposed to gain-framed and loss-framed messages. More specifically, participants exposed to gain-framed messages will report lower levels of medical mistrust compared to those exposed to loss-framed messages.*
**H2.** *There will be a statistically significant difference in PrEP self-efficacy between participants exposed to gain-framed and loss-framed messages, and this relationship will be partially mediated by medical mistrust.*
**H3.** *Identity resilience will be statistically significantly associated with both medical mistrust and PrEP self-efficacy.*
**H4.** *Medical mistrust will be statistically significantly associated with PrEP self-efficacy.*

## 2. Methods

### 2.1. Ethics

A favourable ethical opinion was granted by the Schools of Business, Law, and Social Sciences Research Ethics Committee of Nottingham Trent University. Participants provided informed consent before commencing the study, and they could stop or withdraw at any point during the study or up to four weeks after completion.

### 2.2. Design and Participant Sample

A sample of 253 MSM in the United Kingdom was recruited using convenience sampling, via social media, to participate in this study. Participants aged 18 or above, assigned male at birth, single or in a polyamorous/consensually non-monogamous relationship, comfortable reading and writing in English, and not currently taking PrEP (but eligible to do so) were eligible to participate in the 10 min experiment, which was hosted on the online platform Gorilla (https://gorilla.sc/). All experimental instruments were in English.

This study used a between-subjects experimental design with two levels of message framing: loss-framed and gain-framed vignettes. The outcome variable was PrEP self-efficacy. Medical mistrust was included as a mediating variable, and identity resilience and age were included as covariates. An a priori power analysis using G*Power (version 3.1.9.7) [39] indicated that a minimum of 175 participants would be required to detect small-to-moderate effect sizes across all regression paths in the mediation model with 80% power at α = 0.05. The final sample size, therefore, exceeded this threshold.

Of the 253 participants enrolled in the study, the data of 7 were removed due to ineligibility (e.g., not being assigned male at birth). Therefore, the final sample was *N* = 246. Participants were randomly assigned to one of the two vignette conditions: loss-framed (*n* = 124) or gain-framed (*n* = 122) messaging.

### 2.3. Experimental Manipulation

Two narrative vignettes were developed to manipulate message framing (loss vs. gain) [24], such as: “The NHS encourages its use to increase protection against HIV for everyone” (gain) vs. “PrEP may not prevent other STIs or pregnancy, but it alleviates a lot of stress and worry” (loss). The vignettes were developed specifically for the purpose of this study. The vignettes were in English and designed to be appropriate for a reading age of 9–12 years. Characters were developed for the narrative vignettes and the characteristics of said characters were designed to be as neutral as possible to minimise cultural, socio-economic, or identity-based bias [40,41]. For instance, names such as *Sam* and *Max* were selected for their gender-neutral associations while still allowing for plausible identification as gay men within the narrative. Additionally, characters had no identifiable race or ethnicity, and their occupations and living situations were described in general terms to avoid cues that might influence interpretation based on class, culture, or community affiliation. Each vignette described a character’s PrEP consultation and then their experience of discussing it with a friend. The issues discussed as part of the narrative (e.g., fear of side-effects) were based upon previous study findings [27,42].

Prior to conducting the study, the vignette conditions were piloted to ensure readability and clarity, consistent with best practices in social psychology research [43]. Thirty-one participants read both vignettes and completed a best-practice checklist adapted from Riley and colleagues [44]. Participants did not take part in the main study. There were no eligibility criteria other than being aged 18 years or over and being comfortable reading in English. They rated the vignettes according to the described checklist: each passage on a 5-point Likert scale (1 = strongly disagree to 5 = strongly agree) across several dimensions: narrative coherence, cultural and socio-economic neutrality, character realism, readability, length, structural similarity, and thought-provoking content. All criteria were rated between 4 and 5 on average, indicating high levels of perceived quality and comparability across the vignettes. The only exception was the loss-framed vignette, which received a lower average score of 3/5 for socio-economic neutrality, suggesting it was perceived as slightly more biased in tone or content. Nonetheless, the original vignettes were retained to preserve structural and thematic comparability across conditions, and because the feedback did not indicate fundamental misunderstanding or reduced clarity.

### 2.4. Measures

Participants provided demographic data, namely their age, sexuality (gay, bisexual, other), ethnicity (white, mixed, Asian/British Asian, Black, and other), and relationship status (single, open/polyamorous, other). They indicated whether they had engaged in condomless sexual intercourse in the last 6 months (yes vs. no) and whether they had received a STI diagnosis in the last 6 months (yes vs. no).

The 8-item Perceptions of PrEP Self-Efficacy Behaviour subscale [45] was used to measure PrEP self-efficacy on a 4-point Likert scale (1 = very hard to do to 4 = very easy to do). The scale includes items, such as “How difficult would it be for you to take a medicine like PrEP every day?” Higher scores indicate greater PrEP self-efficacy. This scale has shown good internal reliability in other studies (α = 0.74) [27].

The 10-item Health Care System Distrust Scale [46] was used to measure medical mistrust on a 5-point Likert scale (1 = strongly disagree to 5 = strongly agree). The scale includes items, such as “Some medicines have things in them that they don’t tell you about.” A higher mean score indicates higher medical mistrust. This scale had acceptable internal validity in analogous research (α = 0.79) [47].

The 16-item Identity Resilience Index [48], consisting of four subscales (self-esteem, self-efficacy, continuity, and positive distinctiveness), was used to measure identity resilience on a 5-point scale (1 = strongly disagree to 5 = strongly agree). The scale includes items, such as “I feel unique” (distinctiveness) and “Thanks to my resourcefulness, I know how to handle unforeseen situations” (self-efficacy). A higher mean score indicates higher identity resilience. This scale had excellent internal reliability in related research (α = 0.83) [30].

### 2.5. Statistical Analysis

Sample characteristics and descriptive statistics were analysed using IBM SPSS Version 29. These included means, standard deviations (SD), and internal validity for each variable.

Mediation analyses were conducted using the PROCESS macro for SPSS (Version 4.2; [49]). Analyses were based on 5000 bootstrapping samples with 95% confidence intervals (CI), using the percentile method. Although the data met assumptions of normality, bootstrapping was employed to provide robust estimates of indirect effects, as it does not rely on the assumption of normality of the sampling distribution and is recommended for mediation analyses [50]. Bootstrapped confidence intervals were used to test the significance of indirect effects, with significance inferred if the confidence interval did not include zero.

In the current study, a single mediation model (Model 4 in PROCESS) was tested with PrEP self-efficacy as the outcome variable. Message framing (loss-framed vs. gain-framed) served as the independent variable, and medical mistrust was examined as the mediator. Identity resilience was included as a covariate, given the documented relationship with trust in medical systems and confidence in health behaviours (e.g., [11,30]). Age was also included due to known associations with health attitudes and risk perceptions, including in the context of PrEP (e.g., [27]).

## 3. Results

Participants (*N* = 246) were aged 18–72 years (*M* = 31.60, *SD* = 0.65). The majority of the participants were gay (61.4%), white (82.9%), and single (93.1%). Further sociodemographic characteristics are presented in Table 1. While the study was open to a broad range of MSM, the resulting sample was predominantly white, reflecting the limitations of convenience and online recruitment methods, which can underrepresent racially minoritized groups due to well-documented mistrust in health research and broader systemic exclusions [51].

### 3.1. Descriptive Statistics

Descriptive statistics and bivariate correlation coefficients for the constructs in the theoretical model are presented in Table 2.

### 3.2. Correlations

Identity resilience was significantly and negatively, albeit weakly, correlated with medical mistrust (*r* = −0.20, *p* < 0.001). Identity resilience was also moderately, positively correlated with PrEP self-efficacy (*r* = 0.39, *p* < 0.001). Additionally, medical mistrust was negatively, albeit weakly, correlated with PrEP self-efficacy (*r* = −0.21, *p* < 0.001). See Figure 1 for a visualisation of the correlation coefficients.

### 3.3. Mediation Analysis

To test the hypotheses, a mediation analysis was conducted (see Figure 2). The first model examined the effect of message framing on medical mistrust while controlling for age and identity resilience. All standard regression assumptions (normality, linearity, homoscedasticity, and independence of residuals) were met, and no issues of multicollinearity were identified (all variance inflation factors < 1.5) [52].

The overall model was statistically significant, *F*(3, 242) = 5.13, *p* = 0.002, with *R*^2^ = 0.06, indicating that approximately 6% of the variance in medical mistrust was explained by the predictors. Message framing significantly predicted medical mistrust, *b* = −0.188, *SE* = 0.083, *t* = −2.27, *p* = 0.024, 95% *CI* [−0.351, −0.025], such that loss-framed messages were associated with higher medical mistrust. Identity resilience was also a significant predictor of medical mistrust, *b* = −0.228, *SE* = 0.069, *t* = −3.30, *p* = 0.001, 95% *CI* [−0.365, −0.092]. Age was not a significant predictor, *b* = 0.003, *SE* = 0.004, *t* = 0.63, *p* = 0.528.

The second model examined the effect of message framing and medical mistrust on PrEP self-efficacy while controlling for age and identity resilience. This model was significant, *F*(4, 241) = 12.69, *p* < 0.001, with *R*^2^ = 0.17, indicating that approximately 17% of the variance in PrEP self-efficacy was explained by the predictors. Medical mistrust significantly predicted PrEP self-efficacy, *b* = −0.115, *SE* = 0.047, *t* = −2.45, *p* = 0.015, 95% *CI* [−0.207, −0.023], indicating that higher medical mistrust was associated with lower self-efficacy. However, the direct effect of message framing on PrEP self-efficacy was not significant, *b* = −0.046, *SE* = 0.061, *t* = −0.76, *p* = 0.448, 95% *CI* [−0.167, 0.074]. Identity resilience was a significant positive predictor of PrEP self-efficacy, *b* = 0.309, *SE* = 0.052, *t* = 5.99, *p* < 0.001, 95% *CI* [0.207, 0.410], while age was not significant, *b* = −0.0001, *SE* = 0.003, *t* = −0.04, *p* = 0.966.

The indirect effect of message framing on PrEP self-efficacy through medical mistrust was statistically significant, *b* = 0.022, bootstrapped *SE* = 0.013, 95% *CI* [0.001, 0.052], indicating that message framing influences PrEP self-efficacy indirectly via its effect on medical mistrust.

Overall, the results show a significant indirect effect of message framing on PrEP self-efficacy through medical mistrust, even after controlling for age and identity resilience.

## 4. Discussion

This study aimed to examine two key areas relevant to increasing PrEP uptake among MSM in the UK. First, it investigated whether gain- or loss-framed narratives influence medical mistrust and PrEP self-efficacy. Second, it examined the role of identity resilience in shaping both trust in medicine and PrEP self-efficacy. Hypotheses 1, 3, and 4 that gain-framed messages would reduce medical mistrust compared to loss-framed messages; that higher medical mistrust would be associated with lower PrEP self-efficacy; and that identity resilience would be negatively associated with medical mistrust and positively associated with PrEP self-efficacy were fully supported. Therefore, we can reject the null hypotheses. Hypothesis 2, that message framing would be directly and, via medical mistrust, indirectly associated with higher PrEP self-efficacy was only partially supported, as there was, in fact, only an indirect relationship between message framing and PrEP self-efficacy through the mediation of medical mistrust. Therefore, we can only partially reject the null hypothesis.

The findings are consistent with prior research demonstrating that framing health messages in terms of gains rather than losses can reduce psychological resistance and stigma [26]. This suggests that positively framed, empowering messages may be particularly effective in promoting PrEP uptake among MSM, a group that has historically faced medical stigma and structural barriers [24,34]. The significant, indirect effect of message framing on PrEP self-efficacy via medical mistrust highlights the central role of trust in medicine, a barrier well documented in studies on HIV prevention [9,10,11]. While the indirect effect was modest, it is still practically meaningful given that even small improvements in self-efficacy can influence health decision-making, especially in populations where baseline mistrust is high. This suggests that interventions aiming to shift perceptions of medicine or reduce medical mistrust, even incrementally, can have significant effects on PrEP engagement.

This study builds on existing research by showing that gain-based framing can strengthen future interventions to increase PrEP self-efficacy, which in turn may enhance uptake and adherence [53,54]. Importantly, the experimental methodology employed here is novel, providing the first causal evidence linking gain-framed messaging to key psychological outcomes within the context of HIV prevention among UK-based MSM.

The role of identity resilience extends previous work by Gifford and colleagues [27], demonstrating that MSM with higher identity resilience are less likely to mistrust medicine and are more confident in their ability to use PrEP. This finding reinforces Breakwell’s [28] conceptualisation of identity resilience as a key psychological resource in health behaviour change and highlights its significance in HIV prevention. Pertinently, these findings contribute to the literature describing identity resilience as a trait-like psychological construct that remains relatively stable [55] and is therefore unlikely to be influenced by short-term exposure to messaging or recall effects [56]. Moreover, the results are consistent with the findings of qualitative studies that show that identity resilience (and its constituent elements) functions as both a predictive and protective factor in health behaviour change, including PrEP usage [8,42,57].

These findings have important implications for health communication strategies. Interventions designed to increase PrEP uptake should include gain-framed messages that emphasise empowerment and control to reduce medical mistrust and enhance PrEP self-efficacy. Specifically, among MSM, who remain a key demographic group in HIV prevention, these empowerment-focused, gain-framed messages can directly address barriers, such as mistrust and stigma, providing individuals with a greater sense of control over their health and increasing their confidence in using PrEP [58]. By integrating such positive framing into public health campaigns, organisations can normalise PrEP as a form of self-care and empowerment, reaching a wider audience and reinforcing the notion that taking PrEP is an empowering choice rather than a marker of being “high-risk” [59]. Additionally, interventions should incorporate strategies to build identity resilience, such as fostering positive sexual identity, social support, and a sense of belonging, to further reduce mistrust and stigma [60,61]. The results suggest that such an integrated approach could provide a “double benefit” and simultaneously address psychological barriers, such as medical mistrust, and build confidence in relation to PrEP use.

However, it is also important that such narratives are cognizant of both community and structural barriers [62]. Culturally relevant content that relates to the identities of target populations will help improve the PrEP cascade further and aligns well with narrative, gain-framed techniques [63]. Narrative interventions in non-Western settings that are different from the cultural context in which this study was conducted should be adapted to local cultural norms and address distinct structural challenges (e.g., heightened stigma or legal barriers) through community collaboration [64]. Involving local stakeholders in the design of HIV-prevention narratives has been shown to enhance cultural authenticity and engagement in non-Western settings, while in racially diverse MSM populations, co-produced and culturally tailored messaging is essential to ensure relevance, resonance, and shared ownership across ethnic subgroups [65]. For example, the “*HeHe Talks*” project in Hong Kong developed a web-based HIV-prevention intervention where local MSM shared firsthand video narratives about safer sex, illustrating how health narratives can be co-produced with diverse MSM audiences [64]. Similarly, Daniels and colleagues [66] used participatory role-play narratives in South Africa to improve self-efficacy and healthcare engagement among MSM living with HIV.

Ethical considerations are equally essential, particularly as narrative interventions risk oversimplifying complex health decisions or exerting undue persuasive influence if not carefully designed. Transparency, authenticity, and collaboration with the communities represented are essential to mitigate unintended effects and ensure narratives empower rather than manipulate [67]. These principles align with the World Health Organization’s “*Brief Sexuality-Related Communication*” guidelines [68], which emphasise person-centred communication grounded in human rights, cultural sensitivity, and empowerment rather than coercion. Applying such frameworks helps ensure that narrative-based interventions support autonomy and well-being while upholding professional and ethical standards in public health communication [69].

While the model accounted for 17% of the variance in PrEP self-efficacy, it is important to acknowledge the remaining 83%, which likely reflects the influence of additional psychological, social, and structural factors not captured in the current analysis. Psychological barriers to PrEP self-efficacy do not occur in isolation [7]; rather, they are shaped by broader contexts, including structural barriers, such as limited access to culturally competent or specialist sexual health services, which may constrain individuals’ perceived ability to initiate or maintain PrEP use. Other unmeasured but potentially impactful variables include PrEP-related stigma and internalised shame, which are conceptually distinct from general medical mistrust and have been shown to undermine self-efficacy [70,71]. In addition, misunderstandings or fears about PrEP side effects may reduce confidence in one’s ability to adhere to the regimen [8]. Future research should therefore explore the interplay between these psychosocial factors (e.g., internalised homonegativity, health-related anxiety, and perceived social support) and how they interact with communication strategies to shape PrEP self-efficacy. A more holistic approach that integrates individual, interpersonal, and structural influences may better account for the complexity of PrEP decision-making and improve the effectiveness of targeted interventions.

This study is not without limitations. First, the use of a convenience sample may limit the generalisability of the findings to broader intersections of MSM, trans, and gender-diverse populations and non-Western contexts. Indeed, the sample was predominantly White British and highly educated, and, thus, future research should aim to replicate these findings using a more stratified sample (e.g., including MSM of colour). Second, the narrative vignettes were designed to be culturally relevant and were piloted for clarity and neutrality. However, we acknowledge that no confirmatory factor analysis was conducted for the adapted measures used in this study, nor was the cultural appropriateness of these instruments formally validated for diverse British MSM populations beyond language suitability. This could be enhanced in future work if the vignettes were co-produced with members of specific subgroups of MSM, as it would maximise cultural validity and resonance. Third, while the experimental manipulation of message framing provides support for causal interpretations of that pathway, the cross-sectional design of the mediation analysis limits the ability to draw definitive conclusions about causation. Additionally, narrative interventions may introduce interpretive variability, as individuals engage differently with story-based content, and may be especially susceptible to social desirability bias in self-reported outcomes, such as PrEP self-efficacy. Longitudinal research would provide additional evidence regarding causal relationships between identity resilience, medical mistrust, and PrEP self-efficacy over time.

## 5. Conclusions

In conclusion, this study highlights the importance of effective health communication and psychological resilience in promoting PrEP self-efficacy in MSM. The findings support the use of gain-framed, narrative-based messaging to reduce medical mistrust and enhance PrEP self-efficacy, while also showing the role of identity resilience in fostering trust and confidence. Interventions that integrate these strategies may be effective in addressing the complex psychosocial barriers to PrEP use and advancing public health goals for HIV prevention and PrEP uptake.

## Figures and Tables

**Figure 1 healthcare-13-01981-f001:**
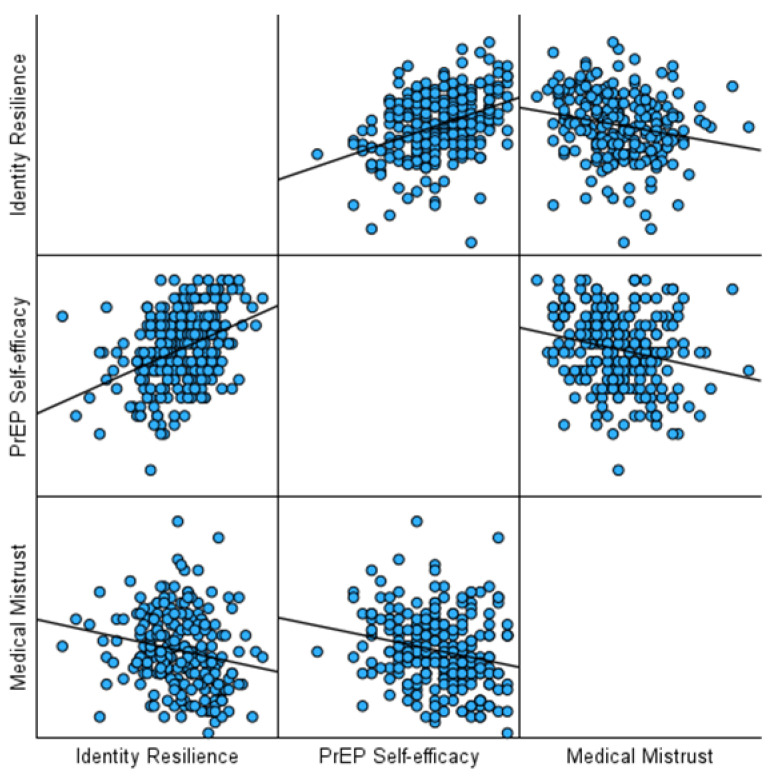
A graph of the correlation coefficients.

**Figure 2 healthcare-13-01981-f002:**
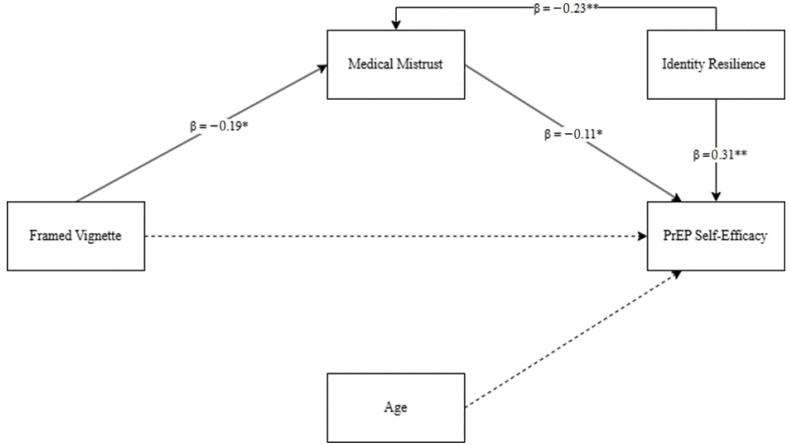
Mediation model predicting PrEP self-efficacy. Note: dashed lines indicate non-significant paths. * *p* < 0.05; ** *p* < 0.001.

**Table 1 healthcare-13-01981-t001:** Sample characteristics of UK MSM (*N* = 246).

Demographic	Number	Percentage
*Sexuality*		
Gay	n = 151	61.4%
Bisexual	n = 92	37.4%
Other	n = 3	1.2%
*Ethnicity*		
White	n = 204	82.9%
Mixed	n = 9	3.7%
British Asian/Asian	n = 20	8.1%
Black	n = 7	2.8%
Other	n = 6	2.4%
*Relationship Status*		
Single	n = 229	93.1%
Open/Polyamorous Relationship	n = 15	6.1%
Other	n = 2	0.8%
*Engaged in condomless sex in the last 6 months*		
Yes	n = 84	34.1%
No	n = 162	65.9%
*Had a STI diagnosis in the last 6 months*		
Yes	n = 7	2.8%
No	n = 239	97.2%

**Table 2 healthcare-13-01981-t002:** Descriptive statistics for all the constructs.

Variable (*N* = 246)	Mean	*SD*	Cronbach’s Alpha (α)	95% Confidence Boundaries
Identity resilience	3.41	0.60	0.84	[0.82 to 0.86]
Medical mistrust	2.62	0.66	0.77	[0.73 to 0.80]
PrEP self-efficacy	3.04	0.52	0.78	[0.75 to 0.81]

## Data Availability

Data is available at reasonable request to the corresponding author.

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
