# Peer review of "Gain-Framed Health Messaging, Medical Trust, and Pre-Exposure Prophylaxis (PrEP) Self-Efficacy: An Experimental Study"

_healthcare, 2025, doi:10.3390/healthcare13161981_

Round 1

Reviewer 1 Report

Comments and Suggestions for Authors

The study fills an important gap by examining the relationship between framing health messages (gain vs. loss) and medical trust in an MSM group with an experimental design to increase PrEP use.

The article can be published with a few corrections.

1) Based on which criteria was the selected sample size decided? What are Type I, II, and Power?

2) If power analysis was not performed, post-hoc power analysis should be reported.

3) The r values ​​obtained in the 3.2. The correlations section generally indicates a weak positive or negative relationship. It would be useful to write it.

4) Adding the graph of correlation coefficients provides visualization.

5) It is recommended to briefly add Model fit indices or assumption checks (multicollinearity) in the applied model.

6) In the tested hypotheses (H1, H2, H3, H4), it is understood from the sentence structure that a one-way hypothesis is established. Hypotheses should be established in two directions.

7) In Model 2, 17% of the variance is explained; what are the other potential variables for the remaining 83%? Should be added to the discussion.

8) It is recommended that the acceptance or rejection of the tested hypotheses be briefly added to the findings.

Author Response

Dear Reviewer 1,

Comment: 1. Based on which criteria was the selected sample size decided? What are Type I, II, and Power?

Reply: Thank you for highlighting this. We had conducted an a priori power analysis across the regression lines to ascertain idea of required sample size. We have now included the following information:

An a priori power analysis using G*Power [39] indicated that a minimum of 175 participants would be required to detect small-to-moderate effect sizes across all regression paths in the mediation model with 80% power at α = .05. The final sample size therefore exceeded this threshold.  (lines177 -180)

Comment: 2. If power analysis was not performed, post-hoc power analysis should be reported.

Reply: Please see above response

Comment: 3. The r values ​​obtained in the 3.2. The correlations section generally indicates a weak positive or negative relationship. It would be useful to write it.

Reply: We have now indicated the strength of the correlations. Please see lines 272 to 276

Comment: 4. Adding the graph of correlation coefficients provides visualization.

Reply: Thank you for this suggestion. We have now included a correlation matrix to help visualization of the coefficients. Please see figure 1. 

Comment: 5. It is recommended to briefly add Model fit indices or assumption checks (multicollinearity) in the applied model.

Reply: The assumption checks for mediation analysis are that of multiple regression and we have now included this as suggested:

The first model examined the effect of message framing on medical mistrust while controlling for age and identity resilience. All standard regression assumptions (normality, linearity, homoscedasticity, and independence of residuals) were met, and no issues of multicollinearity were identified (all variance inflation factors < 1.5) [52]. (lines 281-285)

Comment: 6.  In the tested hypotheses (H1, H2, H3, H4), it is understood from the sentence structure that a one-way hypothesis is established. Hypotheses should be established in two directions.

Reply: Thank you for this comment as suggested, our hypotheses are now two-tailed (i.e., established in two directions).

H1. There will be a significant difference in levels of medical mistrust between participants exposed to gain-framed and loss-framed messages. More specifically, participants exposed to gain-framed messages will report lower levels of medical mistrust compared to those exposed to loss-framed messages.
H2. There will be a significant difference in PrEP self-efficacy between participants exposed to gain-framed and loss-framed messages, and this relationship will be partially mediated by medical mistrust.
H3. Identity resilience will be significantly associated with both medical mistrust and PrEP self-efficacy.
H4. Medical mistrust will be significantly associated with PrEP self-efficacy.

Comment: 7. In Model 2, 17% of the variance is explained; what are the other potential variables for the remaining 83%? Should be added to the discussion.

Reply: We have now expanded our considerations for what other potential variables may influence the variance within the mode. We have incorporated suggestions for future work to address this:

While the model accounted for 17% of the variance in PrEP self-efficacy, it is im-portant to acknowledge the remaining 83%, which likely reflects the influence of addi-tional psychological, social, and structural factors not captured in the current analysis. Psychological barriers to PrEP self-efficacy do not occur in isolation [7]; rather, they are shaped by broader contexts, including structural barriers such as limited access to cul-turally competent or specialist sexual health services, which may constrain individuals’ perceived ability to initiate or maintain PrEP use. Other unmeasured but potentially impactful variables include PrEP-related stigma and internalized shame, which are conceptually distinct from general medical mistrust and have been shown to undermine self-efficacy [68], [69]. In addition, misunderstandings or fears about PrEP side effects may reduce confidence in one’s ability to adhere to the regimen [57]. Future research should therefore explore the interplay between these psychosocial factors, (e.g., inter-nalized homonegativity, health-related anxiety, and perceived social support) and how they interact with communication strategies to shape PrEP self-efficacy. A more holistic approach that integrates individual, interpersonal, and structural influences may better account for the complexity of PrEP decision-making and improve the effectiveness of targeted interventions. (Lines 380 – 396).

Comment: 8. It is recommended that the acceptance or rejection of the tested hypotheses be briefly added to the findings.

Reply: We have now indicated whether we reject or accept the null hypotheses. 

Reviewer 2 Report

Comments and Suggestions for Authors

Many thanks to the authors for their paper.

It is interesting and significant. As this is the first review and the subject matter is different from my area of expertis, I will focus my comments on methodological aspects and the academic robustness of the paper.

1. It is suggested that the introduction delve deeper into the use of narratives as a behavioral intervention in public health.

2. It is important to indicate whether there was cultural or linguistic adaptation in the instruments, as well as whether a confirmatory factor analysis or validation was performed in this specific population.

3. A greater explanation of the operationalization and justification of the theoretical model on which the mediation is based is needed.

4. Although the limitations section notes the underrepresentation of the sample and the bias inherent in convenience sampling, it would be useful to include further reflection on this in the methodology section.

5. Regarding the results, although R² is reported for the models, the magnitude of the practical effect of the findings should be discussed.

6. The discussion does not address in depth the ethical implications or possible unintended effects of manipulating narratives in public health contexts.

7. The limitations of the narrative approach and the potential for social desirability bias in self-reported responses could be further qualified.

8. It would be interesting if the implications expanded the focus of the findings to non-Western or more vulnerable contexts.

9. Regarding the references, it would be appreciated if more studies outside the British context were used to enrich the discussion.

Author Response

Dear Reviewer 2,

Comment: 1. It is suggested that the introduction delve deeper into the use of narratives as a behavioral intervention in public health

Reply: We thank the reviewer for their helpful suggestion. We have addressed this succinctly by introducing wider research emphasising the use of narrative within a public health context:

Furthermore Murphy and colleagues [19] found that a fictional narrative film about cervical cancer led to greater screening uptake than a non-narrative video, with re-portedly greater knowledge and more positive attitudes towards a “Pap test”. Finally, in their anti-smoking study, Igartua and Rodríguez-Contreras [20] found first-person nar-rative testimonials led to greater identification with the protagonist and this emotional connection strengthened the persuasive impact of the message, increasing intentions to quit smoking. These findings position narrative messaging as a widely used and highly impactful strategy across public health interventions, particularly in engaging resistant audiences and promoting behavior change. (Lines 76 – 84)

Comment: 2. It is important to indicate whether there was cultural or linguistic adaptation in the instruments, as well as whether a confirmatory factor analysis or validation was performed in this specific population.

Reply: Thank you for this comment and we agree it is important. As such we have now indicated on line 173 that all experimental instruments were in English (in line with the recruitment strategy). Please also note we have already included the piloting strategy of the vignettes. Additionally, in line with disciplinary standards, all of the psychometric measures were previously validated and strength of internal consistency in wider research has been reported. Additionally, we have reported the Cronbach’s alpha (see table 2) to report the validation conducted for this data set. 

Comment: 3. A greater explanation of the operationalization and justification of the theoretical model on which the mediation is based is needed.

Reply: We have now elaborated on the operationalisation and justification of the theoretical model fo the mediation, including relevant citations to support this: 

This mediation model proposes that message framing influences PrEP self-efficacy via medical mistrust, as loss-framed messages may heighten medical mistrust among MSM, reducing confidence in PrEP use [10], [36], [37]. Medical mistrust is a known barrier to PrEP engagement, and identity resilience entity was included in the model given its theorized role in buffering the effects of mistrust and reinforcing self-efficacy, particu-larly in the face of identity-relevant threats such as stigma or distrust in healthcare [28], [38]. (lines 142-149)

Comment: 4. Although the limitations section notes the underrepresentation of the sample and the bias inherent in convenience sampling, it would be useful to include further reflection on this in the methodology section.

Reply: We entirely agree this is important to reflect on and as such have included the following, however within the methods section where it felt more contextually relevant:

While the study was open to a broad range of MSM, the resulting sample was predom-inantly white, reflecting the limitations of convenience and online recruitment methods, which can underrepresent racially minoritized groups due to well-documented mistrust in health research and broader systemic exclusions [51] (lines 260-264)

Comment: 5. Regarding the results, although R² is reported for the models, the magnitude of the practical effect of the findings should be discussed.

Reply: We thank the reviewer for this suggestion and now have expanded the discussion to specifically discuss implications for future public health interventions:

Specifically, among MSM, who remain a key demographic in HIV prevention, these empowerment-focused, gain-framed messages can directly address barriers like mistrust and stigma, giving individuals a greater sense of control over their health and increasing their confidence in using PrEP [59]. By integrating such positive framing into public health campaigns, organizations can normalize PrEP as a form of self-care and empowerment, reaching a wider audience and reinforcing the notion that taking PrEP is an empowering choice rather than a marker of being “high-risk” [60]. (lines 353-359)

Comment: 6. The discussion does not address in depth the ethical implications or possible unintended effects of manipulating narratives in public health contexts.

Reply: Thank you for this comment and we agree that this is an important issue. However, within the scope of this paper, we felt it would be inappropriate to engage in an extended discussion of the ethics of manipulating public health narratives, as this study did not involve a public-facing intervention. Rather, the narrative content was presented in a controlled (albeit online) experimental context, reviewed and approved by an institutional ethics committee, and designed in accordance with BPS ethical standards for psychological research. We also note in the discussion that loss-framed messaging can incorporate positively valenced content, which may minimise potential risk if adapted for future interventions. While we touch on ethical considerations in the discussion section to guide future research and practice, we agree this remains an important area for further exploration.:  

At the same time, ethical considerations must be acknowledged, as narrative interventions risk oversimplifying complex health decisions or exerting undue persuasive in-fluence if not carefully designed. Transparency, authenticity, and collaboration with the communities represented are essential to mitigate unintended effects and ensure narratives empower rather than manipulate [67]. (lines 375-379)

Comment: 7. The limitations of the narrative approach and the potential for social desirability bias in self-reported responses could be further qualified.

Reply: We have aimed to further qualify the limitation of social desirability bias as follows:Additionally, narrative interventions may introduce interpretive variability, as individuals engage differently with story-based content, and may be especially susceptible to social desirability bias in self-reported outcomes such as PrEP self-efficacy. (lines 407-410)

Comment: 8. It would be interesting if the implications expanded the focus of the findings to non-Western or more vulnerable contexts.

Reply: We agree this would be very interesting and also received a very similar suggestion from another reviewer. As such we have now included the following:

Narrative interventions in non-Western or more vulnerable settings could be adapted to local cultural norms and address distinct structural challenges (e.g., heightened stigma or legal barriers) through community collaboration [65]. Involving local stakeholders in the design of HIV-prevention narratives has been shown to enhance cultural authenticity and engagement in non-Western settings, while in racially diverse MSM populations, co-produced and culturally tailored messaging is essential to ensure relevance, resonance, and shared ownership across ethnic subgroups [66]. (lines 368-374)

Comment: 9. Regarding the references, it would be appreciated if more studies outside the British context were used to enrich the discussion.

Reply: Thank you for this suggestion. The decision to primarily draw on studies from a British context was intended to ground the empirical work within the cultural and structural conditions in which the experiment was conducted. While HIV prevention is a global issue, it is not homogeneous and varies significantly across regions due to differences in healthcare infrastructure, access, and sociocultural factors. However, in line with Comment 2h, we have sought to enrich the discussion by reflecting on implications for non-Western contexts and have included global studies throughout the paper where theoretically relevant. Our aim in doing so is not to be tokenistic, but to acknowledge both the specificity and broader applicability of the findings in a culturally sensitive way.

Reviewer 3 Report

Comments and Suggestions for Authors

Dear Authors,

Thank you for providing me with the opportunity to read this interesting paper. Below, I have listed my comments:

1) In the introduction,  a brief operational definition of medical mistrust could be provided earlier in the section to anchor readers unfamiliar with the term.

2) The concept od identity resillience is well-explained, but repetition could be reduced slightly (lines 102–109) without losing clarity.

3) Line 165: identity resilience and age were included as a covariates, it should be covariates, you can remove a.

4) In the discussion, hypotheses are referred to out of sequence (e.g., Hypothesis 1, 2, and 4... then again Hypothesis 2… which seems like a typo). You may need to double-check whether this is meant to be Hypothesis 3 or if one was skipped.

5) While cultural resonance and limitations regarding demographic skew are mentioned, the practical implications for racially diverse MSM could be expanded. You may wish to discuss  how message design might vary across racial/ethnic subgroups and the importance of co-production in future intervention research.

I hope this feedback is helpful

Author Response

Dear Reviewer 3,

Comment: 1. In the introduction,  a brief operational definition of medical mistrust could be provided earlier in the section to anchor readers unfamiliar with the term.

Reply: Thank you for this suggestion and this has now been included:

Indeed, medical mistrust, defined as the suspicion or lack of confidence in healthcare systems, providers, or medical information, has been linked to lower perceived PrEP self-efficacy [10], [11] (lines 51-53)

Comment: 2. The concept od identity resillience is well-explained, but repetition could be reduced slightly (lines 102–109) without losing clarity.

Reply: We have now amended this section to read as follows:

Furthermore, recent research by Gifford and colleagues [27] extends this perspective by demonstrating that identity resilience (i.e., a construct comprising self-esteem, self-efficacy, continuity, and positive distinctiveness) predicts domain-specific outcomes such as PrEP self-efficacy. It reflects the capacity to maintain a stable, valued sense of self when confronted with threats like PrEP stigma [28], and has demonstrated clear utility in health behaviour contexts. (lines 110-115)

Comment: 3. Line 165: identity resilience and age were included as a covariates, it should be covariates, you can remove a.

Reply: Thank you for spotting this and we have now rectified this spelling error.

Comment: 4. In the discussion, hypotheses are referred to out of sequence (e.g., Hypothesis 1, 2, and 4... then again Hypothesis 2… which seems like a typo). You may need to double-check whether this is meant to be Hypothesis 3 or if one was skipped.

Reply: Again, thank you for highlighting this issue and we have now rectified this mistake. 

Comment: 5. While cultural resonance and limitations regarding demographic skew are mentioned, the practical implications for racially diverse MSM could be expanded. You may wish to discuss  how message design might vary across racial/ethnic subgroups and the importance of co-production in future intervention research.

Reply: Thank you for this suggestion and we received the same suggestion from another reviewer. We have considered these implications carefully. As such we have now included the following:

Narrative interventions in non-Western or more vulnerable settings could be adapted to local cultural norms and address distinct structural challenges (e.g., heightened stigma or legal barriers) through community collaboration [65]. Involving local stakeholders in the design of HIV-prevention narratives has been shown to enhance cultural authenticity and engagement in non-Western settings, while in racially diverse MSM populations, co-produced and culturally tailored messaging is essential to ensure relevance, resonance, and shared ownership across ethnic subgroups [66]. (lines 368-374)

Round 2

Reviewer 1 Report

Comments and Suggestions for Authors

The explanations given by the author can be considered sufficient.

Author Response

The explanations given by the author can be considered sufficient.

Thank you so much for reviewing this again and for allowing us to improve our manuscript. 

Reviewer 2 Report

Comments and Suggestions for Authors

Many thanks to the authors for the corrections made to the text. The improvement is noticeable.
I would like to share some additional brief comments:
1. Considering that no confirmatory factor analysis was performed, nor is the cultural appropriateness of the instrument for diverse British MSM populations (beyond language) discussed, it is necessary to explicitly include a note on construct validity or the limitation of not having evaluated it, given the specificity of the population.
2. Although the discussion is broadened with implications for the design of messages in public campaigns, especially to promote the use of PrEP, the statistical magnitude (e.g., interpretation of indirect β) is not discussed, which would enrich the applied approach.
3.    I understand that you defend your methodological position regarding the risks of simplification or narrative manipulation based on ethical standards, but it would be ideal to briefly link it to ethical frameworks in health communication (e.g., CDC or WHO guidelines) to strengthen the professional perspective.
4. Although relevant international references were incorporated into the discussion, I insist that it would be convenient to mention comparative experiences with other regions of the world where similar interventions have been implemented.
5.    A final editorial review is suggested to eliminate redundancies and dense technicalities.

Author Response

Dear Reviewer 2,

Comment: Considering that no confirmatory factor analysis was performed, nor is the cultural appropriateness of the instrument for diverse British MSM populations (beyond language) discussed, it is necessary to explicitly include a note on construct validity or the limitation of not having evaluated it, given the specificity of the population.

Reply: Thank you for this comment and we have now explicitly stated this limitation as recommended:

However, we acknowledge that no confirmatory factor analysis was conducted for the adapted measures used in this study, nor was the cultural appropriateness of these instruments formally validated for diverse British MSM populations beyond language suitability. (lines 412-415)

Comment: Although the discussion is broadened with implications for the design of messages in public campaigns, especially to promote the use of PrEP, the statistical magnitude (e.g., interpretation of indirect β) is not discussed, which would enrich the applied approach.

Reply: We agree this would be a very useful discussion point and have now added the following:

While the indirect effect was modest, it is still practically meaningful given that even small improvements in self-efficacy can influence health decision-making, especially in populations where baseline mistrust is high. This suggests that interventions aiming to shift perceptions of medicine or reduce medical mistrust, even incrementally, can have significant effects on PrEP engagement. (lines 332-337)

Comment: I understand that you defend your methodological position regarding the risks of simplification or narrative manipulation based on ethical standards, but it would be ideal to briefly link it to ethical frameworks in health communication (e.g., CDC or WHO guidelines) to strengthen the professional perspective.

Reply: Thank you for this helpful comment and we have linked the discussion of ethical frameworks to the WHO’s guidelines around sexuality related health communications (which seemed most appropriate): 

These principles align with World Health Organization’s “Brief Sexuality-Related Com-munication” guidelines [67], which emphasize person-centered communication grounded in human rights, cultural sensitivity, and empowerment rather than persua-sion. Applying such frameworks helps ensure that narrative-based interventions sup-port autonomy and wellbeing while upholding professional and ethical standards in public health communication [68]. (lines 384-389)

Comment: Although relevant international references were incorporated into the discussion, I insist that it would be convenient to mention comparative experiences with other regions of the world where similar interventions have been implemented.

Reply: Thank you for this helpful comment. We have now, succinctly, incorporated evidence demonstrating the broader applicability of narrative messaging within sexual health behaviour change interventions, including among MSM in non-Western settings.

For example, the “HeHe Talks” project in Hong Kong developed a web-based HIV-prevention intervention where local MSM shared firsthand video narratives about safer sex, illustrating how health narratives can be co-produced with diverse MSM au-diences [66]. Similarly, Daniels and colleagues [67] used participatory role-play narra-tives in South Africa to improve self-efficacy and healthcare engagement among MSM living with HIV. (lines 379-384)

Comment: A final editorial review is suggested to eliminate redundancies and dense technicalities.

Reply: Thank you for this and the authors have been through the manuscript, along with recommendations by the editor, to address issues such as these. 

Reviewer 3 Report

Comments and Suggestions for Authors

thank you, authors, for revising the paper accordingly.

Author Response

thank you, authors, for revising the paper accordingly.

Thank you for the constructive review and allowing us to improve our manuscript.